# Proteomics Analysis of Early Developmental Stages of Zebrafish Embryos

**DOI:** 10.3390/ijms20246359

**Published:** 2019-12-17

**Authors:** Kathiresan Purushothaman, Prem Prakash Das, Christopher Presslauer, Teck Kwang Lim, Steinar D. Johansen, Qingsong Lin, Igor Babiak

**Affiliations:** 1Genomics Group, Faculty of Biosciences and Aquaculture, Nord University, 8049 Bodø, Norway; kathiresan.purushothaman@nord.no (K.P.); cpressla@hotmail.com (C.P.); steinar.d.johansen@nord.no (S.D.J.); 2Department of Biological Sciences, National University of Singapore, 14 Science Drive 4, Singapore 117543, Singaporedbslimtk@nus.edu.sg (T.K.L.)

**Keywords:** egg yolk, embryonic development, LC–MS/MS shotgun proteomics, proteome, zebrafish

## Abstract

Zebrafish is a well-recognized organism for investigating vertebrate development and human diseases. However, the data on zebrafish proteome are scarce, particularly during embryogenesis. This is mostly due to the overwhelming abundance of egg yolk proteins, which tend to mask the detectable presence of less abundant proteins. We developed an efficient procedure to reduce the amount of yolk in zebrafish early embryos to improve the Liquid chromatography–tandem mass spectrometry (LC–MS)-based shotgun proteomics analysis. We demonstrated that the deyolking procedure resulted in a greater number of proteins being identified. This protocol resulted in approximately 2-fold increase in the number of proteins identified in deyolked samples at cleavage stages, and the number of identified proteins increased greatly by 3–4 times compared to non-deyolked samples in both oblong and bud stages. Gene Ontology and Kyoto Encyclopedia of Genes and Genomes (KEGG) analysis revealed a high number of functional proteins differentially accumulated in the deyolked versus non-deyolked samples. The most prominent enrichments after the deyolking procedure included processes, functions, and components related to cellular organization, cell cycle, control of replication and translation, and mitochondrial functions. This deyolking procedure improves both qualitative and quantitative proteome analyses and provides an innovative tool in molecular embryogenesis of polylecithal animals, such as fish, amphibians, reptiles, or birds.

## 1. Introduction

Zebrafish have become a prominent and broadly used model system to study developmental biology, neurogenetic disorders, genetics, toxicology, reproduction, pathology, and pharmacology [1,2,3,4,5]. The genome annotation is relatively well developed [6], and the embryonic transcriptome of zebrafish has been characterized in several studies [7,8,9,10,11]. However, knowledge about the comprehensive proteome dynamics during embryogenesis in zebrafish remains elusive.

Proteome in zebrafish is usually investigated in adult organs or tissues [12,13,14,15]. The overwhelming occurrence of vitellogenin yolk proteins is a limiting factor in a polylecithal embryo, such as in zebrafish, as it hinders global identification of less abundant proteins using mass spectrometry-based techniques [4,16]. Proteolytic peptides of yolk proteins can potentially subdue the ionization of the less abundant proteolytic peptides of non-yolk proteins [17,18]. Consequently, abundant yolk proteins can potentially interfere with the identification of cellular proteins, although the degree of such interference is unknown. To reduce the abundance of yolk proteins, deyolking protocols are employed; they have been used in a number of studies on zebrafish embryos and larvae from 3.3 h post-fertilization (hpf) to 7 days post-fertilization (dpf) [16,19,20,21]. In most extensive studies to date, 5267 and 8363 proteins were identified in zebrafish deyolked embryos at 24 hpf [22,23].

So far, all the studies on zebrafish embryonic proteome were conducted on embryos being at a certain developmental advancement, and the information on the early stages, particularly before the maternal-to-zygotic transition (MZT), is missing. Pre-MZT stages of development are characterized by rapid, synchronous cell cycles (cleavages), and the development is driven by maternally-provided factors, including transcriptome and proteome [24]. Therefore, a knowledge of maternal proteome dynamics seems to be essential for understanding the regulation of early embryonic development in zebrafish. We improved the deyolking procedure, allowing the efficient capture and identification of proteins from the onset of development (1-cell stage). The protocol yielded 2 times more identified proteins compared to the non-deyolked counterparts in cleavage stages, and 3–4 times at oblong and bud stages. Also, the protocol caused minimal loss of proteins. Our improved protocol was effective for the subsequent systematic proteomics studies of zebrafish early embryonic development, and it is applicable to studies on other polylecithal animals.

## 2. Results

### 2.1. Efficiency of the New Extraction Protocol

Application of the existing deyolking protocol [16] to zebrafish early embryos requires a considerable amount of embryos to be sampled, yet the representation of low-abundance proteins is reduced (unpublished observation). Therefore, we developed an improved protocol. The major differences are related to the timing and temperature of the dechorionation step, separation of the protein pellet from a liquid fraction, and the wash step (Figure 1).

We compared our protocol to the protocols by Link et al. [16], which were based on 1-step deyolking with or without subsequent washing steps. For a fair comparison, we compared the reference 1-step deyolking procedure without washing [16] to our 1-step deyolking procedure without washing, and the reference 1-step deyolking plus double wash procedure [16] to our 3-step deyolking plus single wash procedure.

Both methods resulted in a reduction of yolk proteins, and the washing steps further depleted the protein content. Nevertheless, our new protocol yielded a larger number of unique proteins from a smaller number of embryos. We obtained approximately 1.7-fold increase in protein concentration per embryo sample when applying the 1-step deyolking process. When using our 3-step deyolking + single wash protocol, the protein yields per embryo sample were 3.1- and 2.5-fold higher (1-cell and high stage embryos, respectively) than those obtained with the reference protocols [16] with 1-step deyolking + double wash (Figure 2A). The effective number of 1-cell stage embryos needed to collect a workable amount of protein (30 µg) was approximately 2 or 3 times lower when using our 1-step deyolking or 3-step deyolking + single wash protocols, respectively; for the high stage embryos, this number of embryos was approximately 2 times lower than that of the respective reference protocols (Figure 2B). Also, compared to our new protocol, less amount of proteins was harvested with the reference protocol (Figure 2B).

### 2.2. Proteome in Deyolked Versus Non-Deyolked Samples

Generally, the amount of extracted total protein per embryo increased with the developmental advancement of the embryo, and the deyolking procedure greatly reduced the protein concentration. However, this reduction was decreasing from over 25-fold in cleavage stages (1-cell, 16-cell, and 32-cell stages) to approximately 15-fold in the bud stage (Table 1).

Analysis of the digested protein samples using the one-dimensional (1D) mass spectrometry (MS)/MS shotgun proteomics approach (1D shotgun) consistently demonstrated that the deyolking procedure resulted in a greater number of proteins being identified (Appendix A). In the non-deyolked samples, the total numbers of proteins identified throughout the developmental stages were relatively consistent, ranging from 338 to 434 proteins identified in the 1-cell and bud stages, respectively. By comparison, the numbers of proteins identified in deyolked samples in all the developmental stages were considerably higher than in the non-deyolked counterparts. In the cleavage stages, these differences were approximately 2-fold, and increased to over 3-fold in the later developmental stages, ranging from 696 to 1687 proteins identified in the 1-cell and bud stages, respectively (Figure 3A, Appendix A). In contrast to the non-deyolked samples, where there was no apparent correlation between the developmental progression and the total number of proteins identified, deyolked samples resulted in a consistent number of proteins identified throughout cleavage stages (1-cell, 16-cell, and 32-cell), which considerably increased in the later developmental stages (Figure 3A). Most of the proteins identified in the non-deyolked samples were also found in the deyolked counterparts (Figure 3A, Appendix A). The number of proteins unique to the non-deyolked samples (that is, not found in the deyolked counterparts) was relatively stable throughout the developmental stages. In contrast, most of the proteins identified in the deyolked samples were unique, meaning that they were not found in the non-deyolked counterparts, and the number of unique proteins apparently increased throughout the embryonic development from the cleavage stages to the bud stage (Figure 3A, Appendix A).

When looking only to the proteins shared between the non-deyolked and deyolked samples, representation of vitellogenin in the deyolked samples was substantially reduced (36–58 times, depending on developmental stage; Appendix A). At the same time, the representation of non-vitellogenin proteins in the deyolked samples was considerably elevated (2–6 times, depending on developmental stage; Appendix A).

Among the 504 proteins unique to non-deyolked samples, most of them were specific to the cleavage stages, and 42 proteins were found in all the developmental stages. By comparison, out of 2129 proteins unique to the deyolked samples, 420 proteins were found in the cleavage stages only, and 266 proteins were found commonly in all the deyolked samples. In contrast to the non-deyolked sample counterparts, a substantial proportion of unique proteins was found in either or both oblong and bud stages. In total, 465 proteins were present in both non-deyolked and deyolked samples across all the developmental stages (Figure 3B, Appendix A).

### 2.3. Functional Annotations of the Proteome

In both non-deyolked and deyolked samples, the identified proteins were substantially involved in metabolic, ribosome, and biosynthesis of secondary metabolite and proteasome pathways, while enrichments specific to sampling protocol and/or developmental stage were found in certain pathways, such as in proteasome, RNA transport, or thermogenesis pathways (Table 2).

Analysis of representation of the identified proteins annotated to functional Gene Ontology (GO) terms revealed multiple processes, functions, and components overrepresented and underrepresented in both non-deyolked and deyolked samples, with some of them specific to the developmental stage (Figure 4, Appendix A).

To distinguish the effect of the extraction protocol (non-deyolked versus deyolked samples) from the biological features (natural representation of proteins at given developmental stage), we used functional annotations of proteins represented in both non-deyolked and deyolked samples from all the developmental stages as a filtering criterion. In this way, shared GO terms were established by: The same proteins identified in samples from both extraction methods (“Shared” dataset); different proteins in both datasets (“unique ND” and “unique DY” datasets) enriching the same terms; or partially the same and partially different proteins (“Shared” and “unique ND”, “Shared” and “unique DY”, and all the three datasets). Whereas, unique GO terms were established exclusively by proteins from either “unique ND” or “unique DY” datasets.

Clearly, the deyolking procedure yielded a considerable number of unique GO terms, which were not annotated with the proteins identified in the non-deyolked samples (Appendix A). The most prominent, developmentally relevant examples included enrichment in: Cellular component organization, RNA splicing, DNA replication, intracellular transport, cell cycle, translational initiation, and mitochondrial organization, transport, and gene expression (biological process); ATP binding, GTP binding, NADH dehydrogenase activity, ribonucleoprotein complex binding, translation initiation factor activity, ribosome binding, and ligase activity (molecular function); chromosome, endoplasmic reticulum, Golgi-associated vesicle, polysome, spliceosomal complex, cytochrome, and mitochondrial ribosome, matrix, and respiratory chains I and II (cellular component). Similarly, underrepresentation in unique GO terms was developmentally relevant, and it included: Cell–cell signaling, chemical synaptic transmission, intracellular signal transduction, and immune response (biological process); DNA-binding transcription factor activity, transcription regulator activity, channel activity, G protein-coupled receptor activity, and kinase activity (molecular function); cell surface, extracellular region, and plasma membrane-bounded cell projection (cellular component). In contrast to the abundance of unique GO terms annotated with the “unique DY” dataset, there were very few unique GO terms associated with “unique ND” dataset, with the most notably enriched terms in molecular function: Carbohydrate binding and endopeptidase regulator activity (Appendix A).

A certain number of proteins was unique for a given developmental stage (that is, identified only in a single developmental stage), in both non-deyolked and deyolked samples. Interestingly, significantly enriched GO terms for these proteins were different for non-deyolked and deyolked samples, in all five developmental stages investigated (Appendix A).

## 3. Discussion

The improved deyolking procedure resulted in a considerably high quantity of the extracted total protein (Figure 2A) We identified 2575 proteins in total. In the study by Link et al. [16], 57 proteins were found, but six of them were not identified, and two proteins had a duplicated ID. We manually retrieved these 50 IDs, and found that 47 (94%) proteins were present in our dataset. Two of the three proteins not found in our dataset were actually *Cyprinus carpio* and *Drosophila melanogaster* proteins, but their possible homologues in zebrafish were missing in our dataset as well. We used 3 times less embryos in our procedure (Figure 2B) than in the reference procedure [16]. Consequently, we were able to conduct the proteomics analysis of zygotic and cleavage stages of zebrafish for the first time. Most of the proteins identified in the cleavage stages were unique to these stages of development (786 out of 1494; Figure 2B). This indicates a massive dynamics of zebrafish developmental proteome. It needs to be noted that the protein sequence database, which we did not use for annotating MS data, does not include the sequences of micro-peptides. Therefore, we cannot determine whether the method is suitable for harvesting very small proteins and micro-peptides.

KEGG analysis showed that ribosome, biosynthesis of secondary metabolites, carbon metabolism, and proteasome pathways were detectable in all the samples (Table 2). Also, a number of GO terms were detected in both deyolked and non-deyolked datasets (Appendix A). Nevertheless, we observed a substantial increase in the number of identified unique proteins in the deyolked samples as compared to the non-deyolked counterparts. Consequently, they enriched a number of developmentally relevant GO terms, such as the cell cycle, mitochondrial organization, and functions or translation initiation, which were not enriched in the non-deyolked samples (Appendix A). These functional terms are essential for the proper growth and development of the early stage of embryos [25,26,27,28] Knowledge of developmentally relevant proteome will aid understanding the regulation of early embryonic development. The underrepresented GO terms in the deyolked samples were mainly related to cellular signaling, transcription, G protein-coupled receptor activity, and cell surface (Appendix A). These terms were not found underrepresented in the non-deyolked samples. In contrast to the significant GO terms found uniquely in the deyolked samples, there were very few unique GO terms associated with “unique ND” dataset (Appendix A); this indicates that the presence of many embryonic proteins is masked due to the high abundance of yolk.

The functional annotation of cleavage stage proteome is concordant with the canonical knowledge of the catabolism, cell cycle, subcellular organization, and the transcriptional quiescence of pre-MZT embryos [24,29]. Moreover, our data suggest active translation-related processes in the very early embryos. Since zygotic transcripts are not produced yet [8], maternally-provided mRNAs [30] were used to produce the translational machinery and perform the translation. Quantitative proteome analysis throughout the development would be needed to determine the extent of this process, though.

Although the dechorionation/deyolking procedure generally resulted in a substantial increase in the number of identified proteins, it also resulted in a loss of certain proteins as compared to the non-deyolked counterparts (Figure 3A), similarly to a study on 5 dpf zebrafish larvae [21]. Most of the previous proteomic studies did not address the problem of protein depletion due to the deyolking process, and they only used deyolked embryos for the analyses [16,21,22]. In the present study, approximately 30% of the proteins at cleavage stages and 12% at oblong and bud stages were not identified after the deyolking (Figure 3, Appendix A). The GO analysis revealed that these lost proteins are involved in a number of biological processes (translation, protein folding, and mitochondrial organization), molecular functions (generation of precursor metabolites and energy), and cellular component (ribosome and cytosol; Appendix A). Moreover, developmental stage-unique proteins enrich GO terms different for non-deyolked and deyolked samples, in all investigated developmental stages (Appendix A). Altogether, our results suggest that deyolked and non-deyolked samples should be analyzed in parallel to extract a reliable information on the proteome in embryonic development.

## 4. Materials and Methods

### 4.1. Fish

The samples were collected at the zebrafish facility of the Nord University, Bodo, Norway. The experimental process and husbandry were performed in agreement with the Norwegian Regulation on Animal Experimentation (The Norwegian Animal Protection Act, No. 73 of 20 December 1974). This was certified by the National Animal Research Authority, Norway, General License for Fish Maintenance and Breeding no. 17.

The maintenance of zebrafish was done using an Aquatic Habitats recirculating system (Pentair, Apopka, FL, USA) and following established protocols [31]. The fish were fed newly hatched *Artemia sp.* nauplii (Pentair) and SDS zebrafish-specific diet (Special Diet Services, Essex, UK) according to the manufacturers’ instruction. The zebrafish used in the experiment were from the AB line.

### 4.2. Sample Collection

Embryos originated from natural spawning and were collected at five developmental stages (Figure 5). Embryo development was monitored and staged according to Kimmel et al. [32]. For each developmental stage, embryo batches were divided into two variants: Non-deyolked and deyolked. The non-deyolked (intact) embryos were promptly snap-frozen in liquid nitrogen and subsequently stored at −80 °C. The deyolked embryo variants went through the process of dechorionation (removal of chorion) and deyolking. Additionally, the 1-cell (0.5 hpf) and high-stage (3.3 hpf) embryos were collected to compare our deyolking protocol with that by [16].

### 4.3. Dechorionation and Deyolking

Embryos were placed in a Petri dish in phosphate-buffered saline (PBS) supplemented with 1.0 mg/mL Pronase (Sigma Aldrich, St. Louis, MO, USA) [31]. The enzymatic digestion of chorion was performed for 5 min at 37 °C with gentle shaking. Embryos were washed a minimum of 5 times with PBS or until all visible chorion fragments were removed.

The dechorionated embryos were processed using our modified protocol with 3-step deyolking and a single wash. The embryos were transferred to 1.5 mL Eppendorf tubes containing 1.0 mL of deyolking buffer (55 mM NaCl, 3.6 mM KCl, and 1.25 mM NaHCO_3_) and were mechanically disrupted by pipetting repeatedly through a 100 µL tip. The content was gently mixed by inverting the tube several times before centrifugation at 13,000 RPM for 1 min at 4 °C. The supernatant containing the yolk was discarded, and the pellet was re-suspended with the deyolking buffer, vortexed, and centrifuged as above. The procedure was repeated two times. After this, the pellet was re-suspended with 10 mM Tris-HCl (pH 7.4), vortexed, and centrifuged as above. The supernatant was discarded and the pellet (deyolked embryos) was snap-frozen in liquid nitrogen and stored at −80 °C. Additionally, for comparison of our protocol with that of [16], the dechorionated embryos at 1-cell and high stage were subjected to two types of deyolking protocols reported by Link et al. [16]: (1) 1-step deyolking, and (2) 1-step deyolking with two additional wash steps.

### 4.4. Protein Extraction

Both intact (non-deyolked) and deyolked embryo samples were lysed by adding 100 µL of sodium dodecyl sulphate (SDS) lysis buffer (1% SDS; Sigma-Aldrich, St. Louis, MO, USA), 0.5 M triethylammonium bicarbonate buffer pH 8.5 (TEAB; Sigma Aldrich), and 1 × Protease Inhibitor cocktail (Thermo Scientific, Rockford, IL, USA)). The tubes were vortexed and incubated at 90 °C for 30 min, then cooled on crushed ice for 5 min. The lysed material was centrifuged at 13,000 RPM for 20 min at 4 °C. The supernatant, containing the proteins, was collected and transferred to a new Eppendorf tube. The total protein concentration was quantified using a Qubit^®^ 3.0 Fluorometer (Invitrogen, Eugene, OR, USA) and the Qubit™ Protein Assay Kit (Invitrogen) according to the manufacturer’s instructions. After the quantification, the samples were freeze-dried (VirTis BenchTop™ K, Warminster, USA) at −80 °C for 18 h before being shipped to the Department of Biological Sciences, National University of Singapore for proteomics analysis.

### 4.5. Polyacrylamide Gel Electrophoresis

One-dimensional gel electrophoresis was performed to check the efficiency of deyolking protocol, as well as to compare the efficiency of our protocol with the previous ones. Approximately equal concentrations of proteins from each sample were supplemented with 2× SDS loading dye. The samples were denatured by incubation at 95 °C for 10 min and then the proteins were separated by SDS gel electrophoresis (4%–20% Mini-PROTEAN^®^ TGX™ Precast Protein Gels, Bio-Rad, Hercules, California, USA) in SDS running buffer for 1 h. Afterwards, the gel was washed with deionized water for 10 min. The gel was stained with Coomassie Blue (Coomassie Brilliant Blue R-250, Bio-Rad) for 20 min, and de-stained with de-staining solution (40% methanol + 10% acetic acid) overnight at room temperature.

### 4.6. Tube-Gel Digestion and Sample Clean up

For each sample, 30 μg of proteins were used for downstream proteomics analyses. The samples were polymerized in a 10% polyacrylamide gel containing 4% SDS and subsequently fixed with a fixing reagent (50% methanol, 12% acetic acid) for 30 min at room temperature. The gel was cut into small pieces (1 mm^3^) before being washed three times with 50 mM TEAB/50 % acetonitrile (*v*/*v*) and dehydrated with 100% acetonitrile. Next, samples were reduced using 5 mM Tris(2-carboxyethyl) phosphine (TCEP) at 57 °C for 60 min, followed by alkylation with 10 mM methyl methanethiosulfonate (MMTS) for 60 min at room temperature with occasional vortexing. The gel pieces were washed in 500 μL of 50 mM TEAB, dehydrated in 500 μL acetonitrile, and re-hydrated with 500 μL of 50 mM TEAB. The final dehydration step was performed using 100 μL acetonitrile. Trypsinization (1.5 μg trypsin) was performed at 37 °C for 16 h. The digested peptides were centrifuged at 6000× *g* for 10 min to collect the supernatant and stored at −20 °C (protocol modified from [17]. The samples were lyophilized and 30 μL of the dissolution buffer (0.5 M TEAB, pH 8.5) was added to each sample.

### 4.7. 1D LC–MS/MS Analysis

The separation of peptides was performed with an Eksigent nanoLC Ultra and ChiPLC-nanoflex (Eksigent, Dublin, CA, USA) in Trap-Elute configuration. The samples were desalted with a Sep-Pak tC 18 μL Elution Plate (Waters, Miltford, MA, USA), and reconstituted using 20 μL of 2% acetonitrile and 0.05% formic acid. Five microliters (μL) of each sample was loaded on a 200 μm × 0.5 mm trap column and eluted on a 75 μm × 15 cm analytical column (ChromXP C18-CL, 3 μm). A gradient formed by mobile phase A (2% acetonitrile, 0.1% formic acid) and mobile phase B (98% acetonitrile, 0.1% formic acid) was used to separate the sample content at a 0.3 μL/min flow rate. The following gradient elution was used for peptide separation: 0–5% of mobile phase B in 1 min, 5–12% of mobile phase B in 15 min, 12–30% of mobile phase B in 104 min, 30–90% of mobile phase B in 2 min, 90–90% in 7 min, 90–5% in 3 min and held at 5% of mobile phase B for 13 min (protocol modified from [33]).

### 4.8. Protein Identification and Quantification

Peptide identification was carried out with the ProteinPilot 5.0 software Revision 4769 (AB SCIEX) using the Paragon database search algorithm (5.0.0.0.4767) and the integrated false discovery rate (FDR) analysis function. The data were searched against protein sequence databases downloaded from UniProt on May 2018 (total 119,356 entries). The MS/MS spectra obtained were searched using the following user-defined search parameters: Sample Type: Identification; Cysteine Alkylation: MMTS; Digestion: Trypsin; Instrument: TripleTOF5600; Special Factors: None; Species: *None*; ID Focus: Biological Modification; Database for 2018_May_uniprot-zebrafish.fasta; Search Effort: Thorough; FDR Analysis: Yes. The MS/MS spectra were searched against a decoy database to estimate FDR for peptide identification. The decoy database consisted of reversed protein sequences from the UniProt zebrafish database. FDR analysis was performed on the dataset and peptides identified with a confidence interval ≥95% were taken into account.

### 4.9. KEGG and Gene Ontology (GO) Functional Pathways Analysis

To analyse functional pathways associated with protein identified from deyolked and non-deyolked samples, KEGG analysis was performed. The FASTA files were submitted to online server “KAAS - KEGG Automatic Annotation Server” (https://www.genome.jp/kegg/kaas/) in order to get KEGG Orthology (KO) assignments [34]. To map KEGG pathways, the obtained KO numbers were submitted to KEGG mapper web server (http://www.genome.jp/kegg/tool/map_pathway2.html) [35].

GO annotation results and pathway of differentially expressed proteins in pairwise comparisons were obtained using Panther (Panther14.0, 2018_04) [36]. The web conversion tool (https://biodbnet-abcc.ncifcrf.gov) was used to convert unmapped UniProt Accession IDs to ZFIN ID. The web tool Biomart was used to convert unmapped ZFIN IDs to Gene stable ID and to manually identify the unmapped IDs by gene names [37]. UniProt was used to identify protein IDs discontinued (deleted) in the 2018_11 release [38].

## 5. Conclusions

We established an effective deyolking procedure for the proteome analysis of the early stages of zebrafish embryos. Elimination of most of the yolk from early stages of embryos significantly enhanced the identification of cellular proteins with LC–MS-based shotgun proteomics analysis. The improved protocol is applicable to low-input material, enabling investigation of the earliest stages of development. Also, we demonstrated that the deyolking procedure results in the depletion of certain parts of the proteome that can be important in embryonic development. Thus, we suggest that both deyolked and non-deyolked samples should be processed in parallel to ensure a reliable coverage of the proteome during the embryogenesis. Our deyolking procedure will improve both qualitative and quantitative proteome analyses throughout embryonic development of polylecithal animals, such as fish, amphibians, reptiles, and birds.

## Figures and Tables

**Figure 1 ijms-20-06359-f001:**
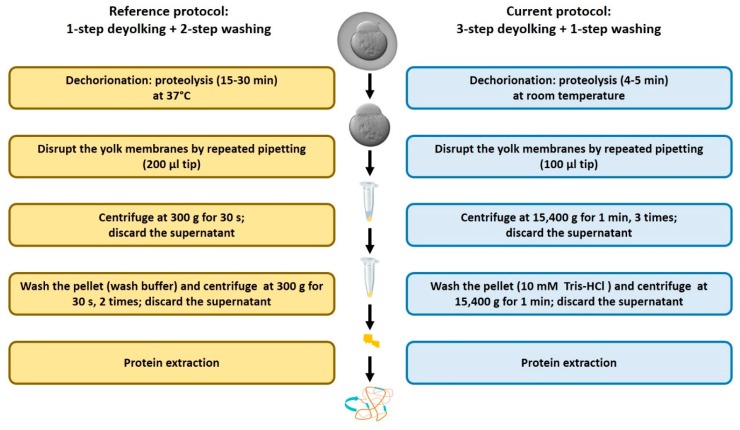
Schematic representation of major differences between the reference [16] and the current deyolking protocols. The detailed information is given in the text.

**Figure 2 ijms-20-06359-f002:**
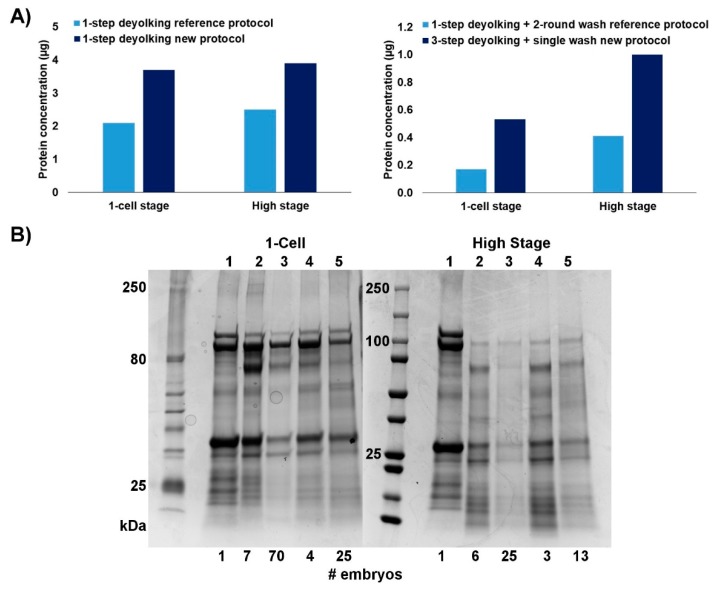
Comparison of the efficiencies of deyolking protocols: reference [16] versus the new one. (**A**) Protein concentration obtained using protocols in 1-step versions (left chart) and in full versions (right chart). (**B**) SDS-PAGE of proteins extracted from zebrafish embryos at 1-cell stage (left panel) and high stage (right panel) using the reference protocols versus new protocols. Lane 1—non-deyolked embryo (control); Lane 2—1-step deyolking reference protocol [16]; Lane 3—1-step deyolking + double wash reference protocol [16]; Lane 4—1-step deyolking (new method); and Lane 5—3-step deyolking + single wash (new method). At the bottom line, number of embryos is given for each sample, from which the proteins were extracted.

**Figure 3 ijms-20-06359-f003:**
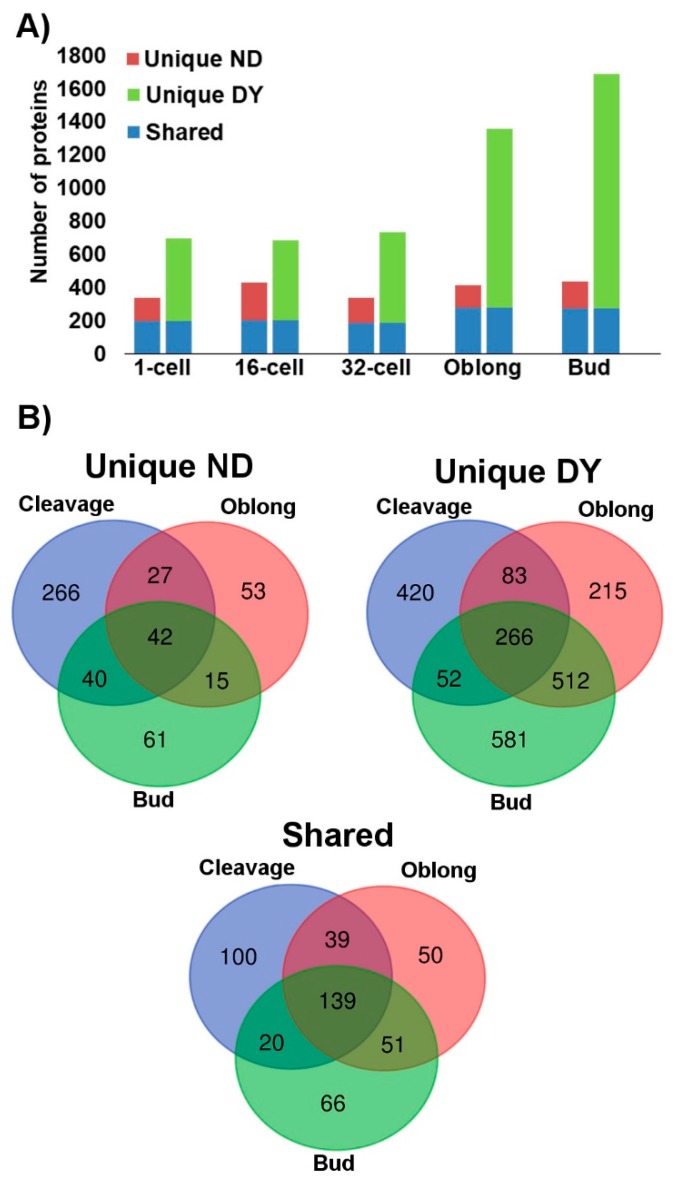
Numbers of proteins identified in samples from intact (non-deyolked, ND) versus deyolked (DY) zebrafish embryos. Unique proteins were found in either ND or DY embryos, whereas shared proteins were found in both ND and DY embryos. (**A**) Total number of unique and shared proteins in ND (left column) and DE embryos (right column) at 1-cell, 16-cell, 32-cell, oblong, and bud developmental stages. (**B**) Specificity and overlap of the identified proteins across the critical stages of early embryonic development: Cleavage stages (1-, 16-, and 32-cell stages combined), maternal–zygotic transition (oblong), and post-maternal–zygotic transition (bud).

**Figure 4 ijms-20-06359-f004:**
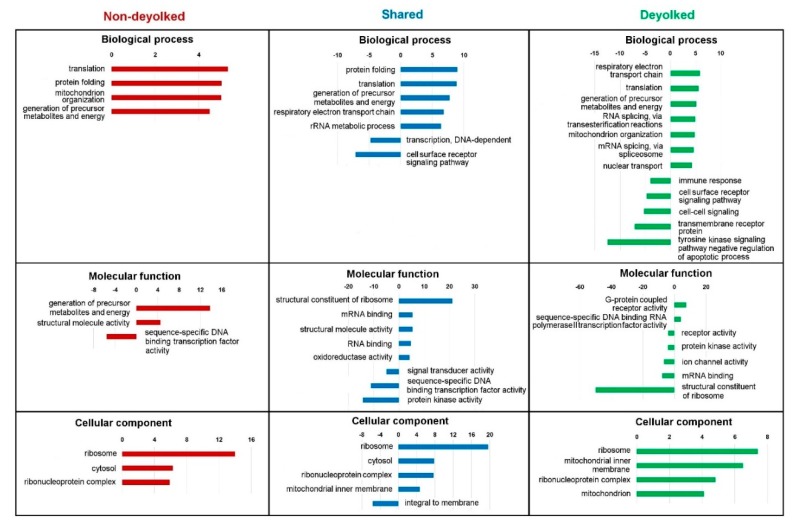
Significantly (false discovery rate (FDR) <0.05) enriched Gene Ontology (GO) terms from SLIM analysis for unique deyolked, unique non-deyolked, and shared proteins grouped by biological process, molecular function, and cellular component. Representation of GO terms containing a minimum 100 reference genes and a fold change ≥4 or ≤4 is given.

**Figure 5 ijms-20-06359-f005:**
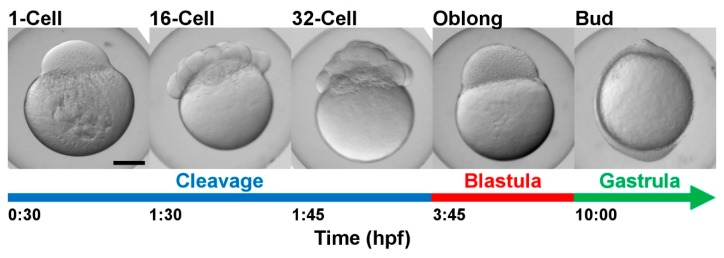
Developmental stages of zebrafish embryos sampled in the present study. hpf = hours post-fertilization at 28.5 °C.

**Table 1 ijms-20-06359-t001:** The amount of protein extracted from deyolked versus non-deyolked embryos.

Developmental Stage	Number of Embryos	Total Sample Volume (µL)	Amount of Extracted Protein (µg)		
			Total	Per µL	Per Embryo
Non-deyolked					
1-cell	28	119	395.08	3.32	14.11
16-cell	20	88	327.36	3.72	16.37
32-cell	40	170	697.00	4.10	17.42
Oblong	20	120	478.80	3.99	23.94
Bud	20	92	524.40	5.70	26.22
Deyolked					
1-cell	575	94	304.56	3.24	0.53
16-cell	300	58	191.41	3.30	0.63
32-cell	400	99	277.22	2.81	0.69
Oblong	225	42	246.54	5.87	1.09
Bud	250	70	413.70	5.91	1.65

**Table 2 ijms-20-06359-t002:** Significant (*p* < 0.05) pathways identified by Kyoto Encyclopedia of Genes and Genomes (KEGG) analysis of proteins from non-deyolked (ND) cleavage, oblong, and bud stage zebrafish embryos, and their deyolked (DY) counterparts. Numbers of proteins mapped to annotated pathways are given.

Pathway Name	ND-Cleavage Stage-Unique	ND- Oblong & Bud Stages Unique	ND-Common in All Stages	DY-Cleavage Stage Unique	DY-Oblong & Bud Stages Unique	DY-Common in All Stages	Shared Proteins
map01100 Metabolic pathways	22	23	6	66	62	87	69
map03010 Ribosome	27	11	2	21	19	11	52
map01110 Biosynthesis of secondary metabolites	12	11	3	21	25	27	32
map04714 Thermogenesis	2	6	1	28	6	37	20
map01200 Carbon metabolism	6	6	1	10	12	21	23
map04141 Protein processing in endoplasmic reticulum	4	0	0	5	14	11	20
map03050 Proteasome	14	4	1	1	15	0	14
map00010 Glycolysis / Gluconeogenesis	6	3	0	2	6	6	12
map00071 Fatty acid degradation	3	1	0	9	1	11	9
map01212 Fatty acid metabolism	8	3	0	7	0	10	8
map04530 Tight junction	5	5	2	6	13	2	7
map03013 RNA transport	7	0	0	3	22	1	6
map04110 Cell cycle	9	2	0	3	12	1	5
map04810 Regulation of actin cytoskeleton	5	1	1	6	12	1	4
map04144 Endocytosis	3	0	0	3	10	6	4
map00230 Purine metabolism	5	5	1	2	11	1	4
map03018 RNA degradation	3	0	0	0	8	1	4
map04210 Apoptosis	0	4	1	4	5	1	4
map00970 Aminoacyl-tRNA biosynthesis	2	1	1	8	12	1	3
map03030 DNA replication	4	3	0	0	14	0	3

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
