# Peer review of "Proteomics Analysis of Early Developmental Stages of Zebrafish Embryos"

_ijms, 2019, doi:10.3390/ijms20246359_

Round 1
Reviewer 1 Report
Kathiresan et al. MS ID: ijms-649252
Proteomics Analysis of Early Developmental Stages of Zebrafish Embryos
In this manuscript, the authors describe the proteome of early zebrafish embryos (up to bud stage) as analyzed in whole intact specimens, and after applying a deyolking protocol developed by the authors for these studies. The experimental design and methodology are overall sound and the results mainly descriptive, providing proteomic data of zebrafish developmental stages that have received little attention in this regard in the past. Therefore, the manuscript will be of general interest to developmental biologists and warrants publication. I am listing below some issues that need clarification, along with some suggestions to increase the manuscript impact.
1.- Regarding the new deyolking protocol, Kathiresan et al. compare their results on total protein yield with those of an earlier protocol published by the Heisenberg lab (Link et al. BMC Dev Biol 2006). It would be illustrative for readers to include a description, albeit succinct, of the motivation and the main differences between the two protocols, at the beginning of the Results section. Moreover, a quantification of the yolk proteins removed after deyolking should be included. This can be easily done comparing the label-free quantification of yolk proteins and cellular proteins in deyolked samples versus non-deyolked samples. One would expect a decrease in the total amount (intensity) of yolk proteins and an increase in the amount of cellular proteins upon deyolking. This should also be illustrated by Western blot analyses of representative proteins (similar to those of Link et al. 2006). Finally, it would be interesting to know how many of the ‘yolk’ proteins removed (partially) after deyolking are indeed yolk constituents, or rather chorionic proteins.
2.- Even though comparing the results from this manuscript with those of Link et al (2006) can be quite difficult when taking into account the differences in analytical sensitivity, it is surprising that only 37 out of the 51 (73%) proteins identified by Link et al. were detected by Kathiresan et al. among the 1355 proteins identified in their analysis. The number seems quite low. Could the authors give a plausible explanation for this?
3.- The authors found a small (but significant) number of cellular proteins not being recovered in deyolked samples. It is unclear to what extent this is due to the new deyolking protocol being too harsh, and whether the ‘old’ protocol published by Link et al. (2006) had similar (or worse) effects.
4.- In Table 2, the number of proteins mapped to some pathways (proteasome, RNA degradation, and DNA replication) found in cleavage-stage embryos in non-deyolked samples is much higher than in deyolked embryos, whereas the opposite is true for these same pathways at later developmental stages. Could this reflect a problem in embryo staging, or do the authors have an alternative explanation?
5.- The analysis of enriched GO terms shown in Fig. 3 is somewhat confusing. Were the data from different developmental stages pooled together? It would be very interesting to analyze whether stage-specific proteins were identified in some stages, but not in others, in deyolked samples versus non-deyolked. For instance, it should be possible to detect the appearance of proteins indicative of the specification of different germ layers in tail-bud stage embryos.
6.- Zebrafish proteome annotation is notoriously poor. Did the authors need to frequently ascribe protein IDs ad hoc and, if so, how was this performed? Spelling mistake in page 10, line 306: …downloaded from UNIPROT…
7.- Page 4, line 114: …, 420 proteins were found in the cleavage stages only, AND 266 proteins…
Author Response
We would like to thank the Reviewer for positive and constructive comments and meaningful suggestions. We have revised the manuscript accordingly, producing a new Figure and three Supplementary Files. This review helped to improve the manuscript. Please find our detailed answers to Reviewer's comments below (in bold font), with references to appropriate changes in the text. We hope the revision is satisfactory.
In this manuscript, the authors describe the proteome of early zebrafish embryos (up to bud stage) as analyzed in whole intact specimens, and after applying a deyolking protocol developed by the authors for these studies. The experimental design and methodology are overall sound and the results mainly descriptive, providing proteomic data of zebrafish developmental stages that have received little attention in this regard in the past. Therefore, the manuscript will be of general interest to developmental biologists and warrants publication. I am listing below some issues that need clarification, along with some suggestions to increase the manuscript impact.
1.- Regarding the new deyolking protocol, Kathiresan et al. compare their results on total protein yield with those of an earlier protocol published by the Heisenberg lab (Link et al. BMC Dev Biol 2006). It would be illustrative for readers to include a description, albeit succinct, of the motivation and the main differences between the two protocols, at the beginning of the Results section. Moreover, a quantification of the yolk proteins removed after deyolking should be included. This can be easily done comparing the label-free quantification of yolk proteins and cellular proteins in deyolked samples versus non-deyolked samples. One would expect a decrease in the total amount (intensity) of yolk proteins and an increase in the amount of cellular proteins upon deyolking. This should also be illustrated by Western blot analyses of representative proteins (similar to those of Link et al. 2006). Finally, it would be interesting to know how many of the ‘yolk’ proteins removed (partially) after deyolking are indeed yolk constituents, or rather chorionic proteins.
The paragraph is added on the study motivation and major differences between the two protocols at the beginning of the Results section (Lines 59-63). Also, a scheme is produced to depict the major differences (Figure 1).
Thank you very much for the suggestion to compare levels of shared proteins between non-deyolked and deyolked samples. We performed the semi-quantitative analysis (emPAI score-based), which shows the rate of depletion of vitellogenins in deyolked samples versus non-deyolked ones. Please see the new Supplementary File 2. In the similar way, we have quantified on the abundances of the non-vitellogenin proteins appearing in both deyolked and non-deyolked datasets, for each developmental stage separately (new Supplementary File 3). Indeed, the majority of these proteins is more abundant in the deyolked samples. The information has been added to the manuscript (Lines 119-123).
In the study by Link et al. (2006), the Western blot has been performed for two proteins only, and its results were not entirely quantitative, although it gave some indication on the quantities of the proteins. Instead, we have provided the semi-quantitative estimation of all the proteins identified. We hope this provides adequate information (Supplementary Files 2 and 3).
Regarding the chorionic proteins, we believe the first step in our deyolking procedure (dechorionation) removes the entire chorion. The chorion is first proteolyzed, then its fragments are washed out, and the washing is repeated several times (minimum 5) until all the remnants are removed.
2.- Even though comparing the results from this manuscript with those of Link et al (2006) can be quite difficult when taking into account the differences in analytical sensitivity, it is surprising that only 37 out of the 51 (73%) proteins identified by Link et al. were detected by Kathiresan et al. among the 1355 proteins identified in their analysis. The number seems quite low. Could the authors give a plausible explanation for this?
We have thoroughly re-analysed the data by Link et al. (2006), and found that some of their original IDs skipped from the comparison. We are sorry we haven’t noticed it in the first instance. We have manually retrieved all their IDs in UniProt. Link et al. report 57 proteins, but 6 of them have not been identified. Out of the remaining 51 proteins, two had duplicated ID. Out of the 50 proteins, 47 (94%) were found in our dataset. Of the three proteins not found in our dataset, one had Cyprinus carpio ID, another one had Drosophila melanogaster ID (we found the closest zebrafish homologues, but they had no match in our dataset), and the third one had a discontinued ID.
We have made appropriate changes in the text (Lines 195-199). Also, for the convenience of the reviewer, we are attaching a working file where we detail the retrieved IDs from the work by Link et al. (2006).
3.- The authors found a small (but significant) number of cellular proteins not being recovered in deyolked samples. It is unclear to what extent this is due to the new deyolking protocol being too harsh, and whether the ‘old’ protocol published by Link et al. (2006) had similar (or worse) effects.
We think the loss of a certain number of proteins from non-deyolked samples, when compared to the deyolked ones, is due to the deyolking process. However, it is unclear whether they are yolk or cellular proteins (or both), which are removed. Also, we don’t know whether this is really a loss, or rather their amounts are lowered below the method’s detectability. Currently, aside from vitellogenins, the yolk-specific protein portfolio in the early development is not well known. We have ongoing study to elucidate this.
We cannot compare this to Link et al (2006) as they did not perform non-deyolked samples in any protein identification level.
4.- In Table 2, the number of proteins mapped to some pathways (proteasome, RNA degradation, and DNA replication) found in cleavage-stage embryos in non-deyolked samples is much higher than in deyolked embryos, whereas the opposite is true for these same pathways at later developmental stages. Could this reflect a problem in embryo staging, or do the authors have an alternative explanation?
We are sure of embryo staging accuracy. Every single embryo has been visually inspected (under microscope) for its developmental stage prior to sampling. Representation of proteins enriching proteasome and RNA degradation KEGG pathways in early stages, higher in non-deyolked than deyolked counterparts, can reflect the actual distribution; from our ongoing research on yolk fraction (unpublished), it looks like both proteasome and RNA degradation processes do occur in the early yolk. Also, in the current dataset, the number of proteins in RNA degradation pathway is very low (3 in NDY, 0 in DY), and the conclusions on potential difference can be erratic. Regarding the DNA replication, it is confusing. These proteins are DNA replication licensing factors MCM 3, 4, 5, and 6 (helicases). Actually, MCM2 and MCM5 are present in the deyolked (DY) cleavage stages, but they did not enrich significantly in the KEGG pathway. Still, more MCM proteins are found in NDY samples. Possible explanation is that the deyolking procedure somehow depleted them. This example shows that both deyolked and non-deyolked samples should be analyzed.
5.- The analysis of enriched GO terms shown in Fig. 3 is somewhat confusing. Were the data from different developmental stages pooled together? It would be very interesting to analyze whether stage-specific proteins were identified in some stages, but not in others, in deyolked samples versus non-deyolked. For instance, it should be possible to detect the appearance of proteins indicative of the specification of different germ layers in tail-bud stage embryos.
Yes, the data in Figure 3 (Figure 4 in the revised version) are pooled. Following the suggestion from the Reviewer, we have produced a new file (Supplementary File 6), where all proteins unique for every developmental stage are listed, separately for NDY and DY protocols. They are given their relative quantification scores. Also, this file contains GO analyses for these proteins stage by stage, separately for the two protocols, as well as the comparison of the enriched GO terms between the two protocols within each developmental stage; only prominent terms (minimum 100 reference genes in a term) and considerable enrichments (minimum 4-fold) were considered for clarity. Interestingly, these enriched GO terms were different for non-deyolked and deyolked samples, in all 5 developmental stages. The text has been added (Lines 188-191 and 239-241).
6.- Zebrafish proteome annotation is notoriously poor. Did the authors need to frequently ascribe protein IDs ad hoc and, if so, how was this performed?
Yes, the annotation of zebrafish proteome is far from ideal. Most of the annotation is automated, and many of the IDs we have found were actually discontinued with UniProt as not validated and dubious. Out of the total 2575 proteins identified in this study, 1564 IDs were annotated with Panther. The remaining IDs were retrieved manually, using multiple resources to identify the sequence and translate the IDs to a form recognizable by the GO tool. It is briefly described in Lines 351-356.
Spelling mistake in page 10, line 306: …downloaded from UNIPROT…
It is corrected now
7.- Page 4, line 114: …, 420 proteins were found in the cleavage stages only, AND 266 proteins…
It is corrected now
Thank you very much for your review.
Reviewer 2 Report
Purushothaman et al describe an improved method for deyolking zebrafish embryos to analyse the proteome. De-yolking of zebrafish embryos at early stages is essential in order to obtain better coverage of the proteins and to minimise contamination from the large yolk mass. A previous study by the Heisenberg group had demonstrated the utility of de-yolking early zebrafish embryos for proteomic analysis (Link et al., 2006). Now, Babiak and coworkers have improved upon the previous method, by incorporating additional washing steps, and show that they can obtain nearly similar coverage upon mass spectrometry analysis using 1/3rd of the sample numbers used by Link et al.
With advancements in mass spectrometry methods, it is now possible to obtain better resolution and depth of coverage of samples by proteomics. Consistently, the study finds a number of classes of proteins in their improved methodology, that are normally under-represented (e.g., splicing factors). The authors also included some additional stages that have not been published by other groups, and therefore, provides some new useful data.
Overall the study is an improvement over previous methods, but some details are either not provided or clearly presented. Some additional information and further details must be provided for the m/s to be suitable for publication.
Major comments:
1) I am puzzled as to why the authors see only 73% of the proteins identified by the Link study. I understand that they are using only 30% of the samples used by Link but the methods for Mass spec have improved considerably since that study. Can the authors increase the sample input slightly and see all or more proteins compared to those reported by Link ? For e.g., would a slightly increased input of 400 embryos instead of 250 make the depth of coverage much better?
2) The authors should provide a step-by-step comparison (flow diagrams) of the Link method versus the Purushottaman protocol to demonstrate the overall ease of the new method and also provide depth of coverage comparison between the methods.
3) In the mass spectrometry analysis, by how much is Vitellogenin representation reduced by the improved method? This information is either not provide or is not easy to find in the current m/s. This would be a clear indication of how "clean" the samples are with the new method compared to the old.
4) Is there a cut-off size below which their method does not detect proteins reliably? For instance, How many micro-peptides/small proteins can they see in their method? Are these proteins under-represented owing to the extra washes? This is question/concern that many readers will have. If there is a an under-representation, the authors should state this so that readers are aware of the limitations, and some idea of the cut off size or range of proteins should be provided.
5) In the discussion, where the authors make comparisons of their proteomic data to the Link and other relevant studies, the authors should also compare to the known RNAseq datasets to see if novel proteins are found.
Author Response
We would like to thank the Reviewer for the positive and constructive comments and meaningful suggestions. We have revised the manuscript accordingly, producing a new Figure and three Supplementary Files. This review helped to improve the manuscript. Please find our detailed answers to Reviewer's comments below (in bold font), with references to appropriate changes in the text. We hope the revision is satisfactory.
Purushothaman et al describe an improved method for deyolking zebrafish embryos to analyse the proteome. De-yolking of zebrafish embryos at early stages is essential in order to obtain better coverage of the proteins and to minimise contamination from the large yolk mass. A previous study by the Heisenberg group had demonstrated the utility of de-yolking early zebrafish embryos for proteomic analysis (Link et al., 2006). Now, Babiak and coworkers have improved upon the previous method, by incorporating additional washing steps, and show that they can obtain nearly similar coverage upon mass spectrometry analysis using 1/3rd of the sample numbers used by Link et al.
With advancements in mass spectrometry methods, it is now possible to obtain better resolution and depth of coverage of samples by proteomics. Consistently, the study finds a number of classes of proteins in their improved methodology, that are normally under-represented (e.g., splicing factors). The authors also included some additional stages that have not been published by other groups, and therefore, provides some new useful data.
Overall the study is an improvement over previous methods, but some details are either not provided or clearly presented. Some additional information and further details must be provided for the m/s to be suitable for publication.
Major comments:
1) I am puzzled as to why the authors see only 73% of the proteins identified by the Link study. I understand that they are using only 30% of the samples used by Link but the methods for Mass spec have improved considerably since that study. Can the authors increase the sample input slightly and see all or more proteins compared to those reported by Link ? For e.g., would a slightly increased input of 400 embryos instead of 250 make the depth of coverage much better?
We are sorry for the confusion. Actually, the major problem was in difficulty to retrieve the IDs of proteins from the study by Link et al; many of these IDs are discontinued, or they refer to discontinued databases. We have manually and thoroughly re-analysed all these IDs. Link et al. report 57 proteins, but 6 of them have not been identified. Out of the remaining 51 proteins, two had a duplicated ID. Out of the 50 proteins, 47 (94%) were found in our dataset. Of the three proteins not found in our dataset, one had Cyprinus carpio ID, another one had Drosophila melanogaster ID (we found the closest zebrafish homologues, but they had no match in our dataset nevertheless), and the third one had a discontinued ID. We have made appropriate changes in the text (Lines 195-199). Also, for the convenience of the Reviewer, we are attaching a working file where we detail the retrieved IDs from the work by Link et al. (2006).
We have used only a portion of the total extracted proteins for the shotgun analysis. The increase of the sample size would not help in sensitivity. Perhaps, if we performed 2D-LCMS with 1st-dimension pre-fractionation, it could help to improve the coverage.
2) The authors should provide a step-by-step comparison (flow diagrams) of the Link method versus the Purushottaman protocol to demonstrate the overall ease of the new method and also provide depth of coverage comparison between the methods.
The paragraph is added on the major differences between the two protocols at the beginning of the Results section (Lines 59-63). Also, a scheme is produced to depict the major differences (Figure 1). We have compared the input material and the yield between the two methods (Figure 2); we obtained 2.5 – 3.1 times more protein from an embryo with the new method (Lines 76-84).
3) In the mass spectrometry analysis, by how much is Vitellogenin representation reduced by the improved method? This information is either not provide or is not easy to find in the current m/s. This would be a clear indication of how "clean" the samples are with the new method compared to the old.
The reduction of vitellogenins is 36-58 times in the deyolked samples, when compared to the non-deyolked counterparts, depending on the developmental stage. We have produced a new Supplementary File 2 providing the details. Also, the information has been added to the manuscript (Lines 119-123). However, we are not able to compare it to the Link et al. (2006) data, because they did not perform protein identification in non-deyolked samples.
4) Is there a cut-off size below which their method does not detect proteins reliably? For instance, How many micro-peptides/small proteins can they see in their method? Are these proteins under-represented owing to the extra washes? This is question/concern that many readers will have. If there is a an under-representation, the authors should state this so that readers are aware of the limitations, and some idea of the cut off size or range of proteins should be provided.
We do not have a definitive cut-off size. In the protein extraction step, actually our method reduces washes to absolute minimum (1 wash), while the original method by Link et al. requires 3 washes. In the analysis step, we used tube-gel as the digestion method. We might lose the micro-peptides/small proteins during the gel washing steps. Additionally, the protein sequence database which we have used for annotating MS data does not include the sequences of micro-peptides. We identified only proteins with mass greater than 2807. There is a number of small proteins in the dataset (please see Supplementary File 1), but we cannot determine whether they are underrepresented or not. We have added text (Lines 204-206) addressing this issue.
5) In the discussion, where the authors make comparisons of their proteomic data to the Link and other relevant studies, the authors should also compare to the known RNAseq datasets to see if novel proteins are found.
All the proteins identified in the current study are annotated with UniProt database. We examined them against the Zebrafish Expression Atlas database (White et al, eLife 2017;6:e30860) and they are matching transcripts.
Thank you very much for the review.
Round 2
Reviewer 2 Report
The revised /m/s is improved and addresses thence concerns I raised.